# Genome-Wide Analysis of the MYB Transcription Factor Superfamily in *Physcomitrella patens*

**DOI:** 10.3390/ijms21030975

**Published:** 2020-02-01

**Authors:** Xiaojun Pu, Lixin Yang, Lina Liu, Xiumei Dong, Silin Chen, Zexi Chen, Gaojing Liu, Yanxia Jia, Wenya Yuan, Li Liu

**Affiliations:** 1State Key Laboratory of Biocatalysis and Enzyme Engineering, Hubei Collaborative Innovation Center for Green Transformation of Bio-Resources, Hubei Key Laboratory of Industrial Biotechnology, School of Life Sciences, Hubei University, Wuhan 430000, China; puxiaojun@mail.kib.ac.cn (X.P.); wyyuan@hubu.edu.cn (W.Y.); 2Key Laboratory for Economic Plants and Biotechnology, Germplasm Bank of Wild Species, National Wild Seed Resource Center, Kunming Institute of Botany, Chinese Academy of Sciences, Kunming 650201, China; rattan@mail.kib.ac.cn (L.Y.); lylyna@foxmail.com (L.L.); dongxiumei@mail.kib.ac.cn (X.D.); chensl1216@163.com (S.C.); chenzexi@mail.kib.ac.cn (Z.C.); liugaojing2012@163.com (G.L.); jiayanxia@mail.kib.ac.cn (Y.J.)

**Keywords:** circadian clock, domain shuffling, MYB, *Physcomitrella patens*, stress responses, tandem gene duplication, block duplication

## Abstract

MYB transcription factors (TFs) are one of the largest TF families in plants to regulate numerous biological processes. However, our knowledge of the MYB family in *Physcomitrella patens* is limited. We identified 116 MYB genes in the *P. patens* genome, which were classified into the R2R3-MYB, R1R2R3-MYB, 4R-MYB, and MYB-related subfamilies. Most R2R3 genes contain 3 exons and 2 introns, whereas R1R2R3 MYB genes contain 10 exons and 9 introns. N3R-MYB (novel 3RMYB) and NR-MYBs (novel RMYBs) with complicated gene structures appear to be novel MYB proteins. In addition, we found that the diversity of the MYB domain was mainly contributed by domain shuffling and gene duplication. RNA-seq analysis suggested that MYBs exhibited differential expression to heat and might play important roles in heat stress responses, whereas CCA1-like MYB genes might confer greater flexibility to the circadian clock. Some R2R3-MYB and CCA1-like MYB genes are preferentially expressed in the archegonium and during the transition from the chloronema to caulonema stage, suggesting their roles in development. Compared with that of algae, the numbers of *MYBs* have significantly increased, thus our study lays the foundation for further exploring the potential roles of *MYBs* in the transition from aquatic to terrestrial environments.

## 1. Introduction

The transcriptional regulation of gene expression is critical for the survival and developmental transitions of an organism, since development and stress responses are often associated with differential gene expression [1,2,3]. The expression of these genes is controlled by transcription factors (TFs), which regulate transcriptional networks to activate or repress gene expression in response to internal and external stimuli [1]. Therefore, deciphering the molecular functions of TFs provides insight into the regulation of gene expression and stress responses. MYB proteins, one of the largest TF families in plants, play important roles in plant growth, development, and stress responses [3,4,5,6,7,8,9,10,11]. The genes encoding MYB domain-containing proteins have been extensively studied since their discovery in the avian myeloblastosis virus (AMV) genome approximately 35 years ago, followed by their identification in *Zea mays* [12,13]. However, the functions of many MYB TFs remain elusive. 

MYB TFs are characterized by a highly conserved MYB DNA-binding domain at their N-terminus. This domain is approximately 50–53 amino acids in length and typically contains one to four imperfect MYB repeats (R0, R1, R2, and R3) [14]. Each repeat adopts a highly similar folding conformation comprising three helices, with three regularly spaced tryptophan residues forming a hydrophobic core. The second and third helices of each repeat form a helix-turn-helix structure [3,15]. The third helix in each repeat functions as a recognition helix that binds to the major groove of DNA at the recognition site C/TAACG/TG [15,16]. The C-terminus of MYB proteins function as activation domains, with considerable variability among MYB proteins, which contributes to the diverse regulatory roles of these TFs [14]. 

Plant MYB proteins are classified into four different types based on the number of repeats in the MYB domain: R2R3-MYB, R1R2R3-MYB, 4R-MYB, and MYB-related proteins [3,14]. R1R2R3-MYB (3R-MYB) proteins predominate in animals, but in the plant kingdom, R2R3-MYB (2R-MYB) proteins are more common [3,17]. In *Arabidopsis thaliana*, 2R-MYB TFs are involved in regulating secondary metabolism [14,18,19,20,21] and plant-specific developmental processes, such as epidermal cell patterning [22,23], sperm cell differentiation, and the cell cycle [24,25]. The 3R-MYB TFs regulate various aspects of the cell cycle [14,26,27,28,29,30]. For instance, some 3R-MYB TFs help maintain the cell cycle by directly repressing the expression of G2/M-specific genes in plants during normal development [28,29,30], thereby regulating plant development. The 4R-MYB TFs, representing the smallest subfamily of MYB TFs, contain four R1/R2-like repeats [3]. Less is known about the functions of 4R-MYB TFs compared to the other MYB TF subfamilies. MYB-related proteins, including CIRCADIAN CLOCK ASSOCIATED1 (CCA1)-like, telomeric DNA-binding protein (TBP)-like, I-box-binding factor-like, and R-R-type [3,31,32], are a heterogeneous collection of several subgroups and typically contain a single or partial MYB repeat [12]. These MYB-related proteins also play diverse roles in plant growth and development, contributing to processes, such as epidermal cell patterning and leaf senescence [22,33], the regulation of circadian rhythms [34,35,36,37,38,39], and stress responses [3,40]. 

*Physcomitrella patens*, a representative of the earliest land plants, occupies a key phylogenetic position and therefore serves as a model system for studying the evolutionary mechanisms associated with the conquest of land, since it experienced the transition from aquatic to terrestrial environments [41]. Unlike most flowering plant genomes, *P. patens* has a unique genome organization, with gene- and TE-rich regions dispersing evenly along the chromosomes, and no TE-rich pericentromeric and gene-rich distal areas were detected [42]. There is evidence that activation of TE during the life cycle of land plants might be linked to its haploid-dominant life style and motile gametes [42,43]. The complexity of multicellular organism evolution is considered to be associated with an increase in the number of regulatory proteins and the expansion of TF gene families [2]. It remains to be determined whether the adaptations associated with plant terrestrialization and the morphological complexity of *P. patens* are tightly linked to the expansion of TF gene families. In *P. patens*, genes encoding enzymes that catalyze chlorophyll and carotenoid biosynthesis as well as cytochrome P450 enzymes have undergone duplication, making the metabolic pathways in this plant more complex than those in the green alga *Chlamydomonas reinhardtii* [41]. In a recent study, Hisanaga et al. (2019) identified a central module consisting of an R2R3-MYB (MpFGMYB) and an antisense RNA that regulates sexual dimorphism in the liverwort *Marchantia polymorpha*, with loss-of-function *Mpfgmyb* mutants exhibiting a conversion of the female-to-male phenotype [44]. Intriguingly, orthologs of MpFGMYB in the flowering plant *Arabidopsis thaliana* have been shown to be expressed in embryo sacs and promote their development [44,45], suggesting that this R2R3-MYB might have a conserved role in regulating female gametophyte development. The colonization of land was considered as a milestone event in the evolution of plants from aquatic to terrestrial; this transition was hypothesized to be facilitated by acquisition of the flavonoid pathway because it provides plants protection against various abiotic stresses, such as drought, extreme temperature fluctuations, and UVB irradiation [21,46]. In angiosperms, R2R3MYB TFs typically act as part of an R2R3MYB-bHLH-WDR (MBW) complex to regulate the biosynthesis of flavonoid [47,48]. The recent finding of MpMyb14, an R2R3MYB, from *M. polymorpha* able to activate flavonoid production in response to abiotic stress [21,49], suggests an early evolutionary adaption to abiotic stresses during land colonization might be achieved by co-option of MYBs and the flavonoid pathway. Thus, unraveling the roles of MYB TFs in land plants will provide insights into plant adaptation to terrestrial environments. In a previous study, Du et al. (2013) identified only 27 MYB-related proteins in the *P. patens* genome [32], whereas Feller et al. (2011) estimated a total of 180 MYB proteins in the *P. patens* genome [2]. Therefore, while the MYB gene family has been characterized in detail in various plant species, little is known about the exact numbers and types of this gene family in *P. patens*. To date, *CCA1/LHY*, which is a key regulator of circadian rhythms, is the only MYB gene to have been functionally characterized in *P. patens* [37]. Therefore, we performed a genome-wide analysis of the MYB gene family in *P. patens*. The results of this study provide a basis for further exploring the functions of this important gene family in land plants. 

## 2. Results

### 2.1. Genome-Wide Identification and Classification of MYB Family Members in P. patens

To identify MYB proteins in *P. patens*, we firstly performed an all-to-all BLASTp analysis against the *P. patens* genome using *Arabidopsis* MYB proteins as query sequences. A total of 101 putative MYB proteins were identified. We also downloaded the sequences of 368 MYB proteins in *P. patens* from the PlantTFDB 4.0 database (http://planttfdb.cbi.pku.edu.cn/). After removing redundant proteins, a total of 120 candidate MYB proteins from *P. patens* were identified. We then examined the candidate MYB proteins for the presence of specific MYB signatures; four of the candidates lacked a MYB signature, leaving 116 putative MYB proteins. 

Next, we performed a phylogenetic analysis of the proteins using the neighbor-joining (NJ) and maximum likelihood (ML) methods. The tree topologies generated by these methods were largely similar, and only a few gene relationships varied between the trees produced by these methods (Figure 1 and Appendix A). 

In the NJ trees, proteins of the same type and with similar domains tended to form groups. These 116 MYB proteins were classified as follows: R2R3-MYB family (50 proteins), R1R2R3-MYB (3 proteins), 4R-MYB (1 protein), and MYB-related (62 proteins) subfamilies (Figure 1a and Appendix A). 

These results suggest that *R2R3-MYB* genes are common in *P. patens*, as in other plants. Based on the classification used by Chen et al. (2006) [50] and our phylogenetic analysis, we further divided the 62 MYB-related proteins into several subgroups: CCA1-like, TBP-like, TRF-like, atypical MYB proteins, and other MYBs lacking typical MYB characteristics.

Both the TBP-like and CCA1-like subgroups, containing 13 and 11 genes, respectively, form three clades (Figure 1a). Although divergence was detected among individual MYB domains, the consensus sequences of the LKDKWRN and SHAQK(Y/F)F motifs are highly conserved in the TBP-like and CCA1-like subgroups, respectively (Figure 2a,b). Furthermore, we identified classes of novel 3R-MYB and 1R-MYB proteins, designated as N3R-MYB and NR-MYB, which only occur in *P. patens* (Figure 1). These proteins harbor unique MYB repeats characterized by long amino acid sequences and complicated gene structures, with many exons and introns. Although I-box-binding factor-like proteins, which belong to the MYB-related subfamily, are present in many angiosperms [32], we did not identify these proteins in *P. patens* (Figure 1a), suggesting that this subgroup likely evolved recently in angiosperms.

### 2.2. Structural Analysis of the MYB Family in P. patens

We investigated the intron–exon structures of the MYB genes in *P. patens*. Although both intron-containing (~86%) and intron-less (~14%) MYB genes are present in *P. patens*, R2R3-MYB and MYB-related genes are the most common MYBs in *P. patens*. It appears that three exons interrupted by two introns is the most common structure for R2R3-MYB genes, accounting for 60% (30) of the total R2R3-MYBs. By contrast, only approximately 14% (9) of MYB-related genes contain three exons and two introns, although such genes predominantly occur in *P. patens*. All 3 R1R2R3-MYB genes in *P. patens* contain 10 exons and 9 introns (Figure 1). Furthermore, all tandem duplicated genes from the R2R3-, R1R2R3-, and CCA1-like subgroups retained the same intron–exon organization, whereas some tandemly duplicated genes from the novel 3R-MYB subgroup had experienced structural divergence, displaying different numbers of exons and introns (Figure 1). Even the expression patterns of tandemly duplicated genes with the same intron–exon organization, such as Pp3c9_19080 and Pp3c9_19090, sometimes differ due to divergence in their regulatory regions. These results suggest that the evolution of gene structure is partly derived from structural divergence in both the coding and regulatory regions (Appendix A).

### 2.3. Domain Shuffling has Contributed to the Diversity of MYB Gene Structures and Motifs

MYB TFs typically have a highly conserved DNA-binding domain (MYB domain) at the N-terminus, and a variable C-terminus [2,3]. To analyze the features of MYB protein sequences, we performed multiple sequence alignment and created sequence logos for R2-, R3-, TBP-like-, and CCA1-like proteins from *P. patens*. The MYB domains of R2, R3, TBP-like, and CCA1-like TFs share significant similarity with those of *Arabidopsis* (Figure 2 and Appendix A). 

Notably, 3, 9, and 13 CCA1-like MYB proteins are present in *Klebsormidium flaccidum*, *Sphagnum fallax*, and *P. patens*, respectively (Appendix A), indicating that these clock-related genes underwent significant expansion upon the colonization of land. In addition to MYB domains, other domains, such as the DnaJ-domain, mediator subunits, and RSC8 domains, were also present in the MYB proteins. These observations suggest that domain shuffling, a process through which exons from different genes are brought together to form new exon–intron structures, might have contributed to the diversity of MYB gene structures and motifs. 

### 2.4. PpMYB Family Genes are Distributed Non-Uniformly on Different Chromosomes and Have Different Subcellular Localizations

The chromosome distribution map (Figure 3) shows that 114 of the MYB genes are dispersed across the 25 chromosomes of *P. patens*, with the gene numbers per chromosome ranging from 1 to 9. Chromosomes 4, 11, and 12 contain the highest number of MYB genes (9 each), accounting for 7.8% of the total, whereas no MYB genes were found on chromosome 22 and 27. This result suggests that MYBs are non-uniformly dispersed across the chromosomes of *P. patens*.

TFs are generally composed of at least two discrete domains, a DNA-binding domain and a transcription activation domain [50], which function together to regulate a variety of physiological, biochemical, and developmental processes by activating or repressing a set of specific genes associated with development or stress responses [1]. Most TFs harbor a nuclear-localization signal, which targets the protein to the nucleus. Given that domain shuffling has contributed to the diversity of MYB proteins, we examined whether all MYBs are localized to the nucleus in *P. patens*. To explore this possibility, we investigated the localizations of several representative MYBs by fusing full-length MYB CDS without stop codon to N-terminus of the green fluorescent protein (GFP) coding sequence using primers listed in Appendix A. As shown in Figure 4 and Appendix A, some DnaJ-containing MYBs are not localized to the nucleus, supporting the notion that not all PpMYBs localize to the nucleus. 

MYB genes that have undergone different patterns of domain shuffling might encode proteins with different localizations, thus contributing to the flexibility of the MYB protein function. 

### 2.5. The Diverse Expression Patterns of MYB TFs Have Important Functional Implications for Development and Heat Stress Responses 

To explore the roles of each type of MYB gene, we retrieved the temporal and spatial expression patterns of MYB genes from the BAR database using the Physcomitrella eFP Browser (Figure 5a,b and Appendix A). 

The R2R3-MYB genes are expressed in various developmental stages (such as the protonema and gametophore stages) and in a range of tissues (such as protoplasts, spores, sporophytes, and archegonia; Figure 5a,b and Appendix A). Eight R2R3-MYB genes (8% of the total) exhibited the highest expression levels in archegonia, and six (12%) showed contrasting expression patterns during the transition from chloronema to caulonema, suggesting that these R2R3-MYBs might play a role in reproduction and in the chloronema-to-caulonema transition. Approximately 6 of the 13 CCA1-like MYB genes (46%) showed contrasting expression patterns during the transition from chloronemata to caulonemata. Some CCA1-like MYB genes showed preferential expression patterns, with higher expression levels in gametophores (Pp3c15_23110) and lower expression levels in rhizoids (Pp3c12_14300 and Pp3c4_5410) (Figure 5a,b and Appendix A). The preferential expression of these CCA1-like MYB genes in specific organs and at specific developmental stages suggests that these clock genes might function in a specific developmental program. Like CCA1-like *MYBs*, some TBP-like *MYBs* were preferentially expressed in rhizoids (Figure 5a,b and Appendix A), pointing to their possible roles in rhizoids. In addition, members of the 3R-MYB-related subgroup were preferentially expressed in archegonia and at the chloronema stage. 

MYBs function in a wide variety of biotic and abiotic stress responses [51,52,53,54]. To gain further insight into the roles of MYBs in stress responses, we examined the transcriptional profiles of MYB genes in *P. patens* subjected to heat stress via RNA-seq. The heatmap produced based on these results divided these genes into four clusters (Figure 5a,b). Cluster I contained 33 members, 28 of which were R2R3-MYBs. Cluster II included seven members, all of which were induced by heat stress. Cluster III included 13 members, 4 of which belonged to the N3R-MYB subgroup. Cluster IV included 35 members, all of which were less strongly induced by heat stress than the other MYB genes examined (*p* < 0.05). 

To explore the underlying mechanisms by which MYB genes are induced by heat stress, we identified the conserved motifs in these genes using the MEME tool. Not all phylogenetically related MYB genes shared similar induction patterns. By contrast, genes with similar cis-regulatory elements and motifs or associated with a specific developmental program tended to cluster together (Figure 5a,b, and Appendix A). 

### 2.6. Block Duplication and Tandem Gene Duplication Contributed Differentially to the Expansion of Different MYB Subgroups 

The increasing complexity of multicellular organisms is thought to be tightly associated with the expansion of gene families encoding TFs [2]. Gene duplication is thought to contribute to the expansion and evolutionary novelty of specific gene families in plants [55]. To explore the extent to which gene duplication contributed to the evolution of the MYB gene family, we analyzed the occurrence of tandem duplication during its evolution. Whereas 16% and 44% of R2R3-MYB genes have undergone tandem and block duplication, respectively, 66.7%, 30.7%, and 85.7% of genes in the R1R2R3-, CCA1-like-, and N3R subgroups, respectively, have experienced tandem duplication (Figure 1 and Appendix A). Therefore, block duplication appears to have been primarily responsible for the expansion of the R2R3 MYB subgroup of genes, whereas tandem duplication contributed greatly to the expansion of genes in the R1R2R3-, CCA1-like-, and N3R subgroups. 

## 3. Discussion

MYB TFs constitute one of the largest TF families in the plant kingdom. These TFs play a wide variety of roles in plant growth, development, and stress responses [21,22,33,52,53]. However, the roles of most MYB genes in *P. patens* have remained largely unexplored. The exact number of MYB genes reported in the *P. patens* genome has varied among studies [2,3,32] due to the lack of a systematic analysis. Here, we performed a systematic characterization of MYB gene structures, analyzed phylogenetic relationships among the identified TFs, and examined conserved motifs, chromosomal locations, tissue- and developmental stage-specific expression patterns, and expression patterns in response to stress. Our findings provide novel insights into the roles of MYB genes in development and heat stress responses. To survive and overcome the restrictions posed by their sessile lifestyle, plants have evolved large families of TFs that coordinate the expression of numerous genes, thereby regulating growth and development [1,56]. MYBs represent one of the largest TF families in plants, accounting for ~9% and 10% (116/1136) of the total number of TFs in *A. thaliana* and *P. patens,* respectively [1]. The number of MYB TFs increased substantially; fewer than 50 are present in green algae, whereas over 100 existed in early land plants [2], suggesting that MYB TF genes might first have undergone large-scale expansion in bryophytes (Figure 1). It was suggested that the *P. patens* genome evolved via at least two rounds of whole-genome duplications (WGDs) [42], and that the process by which seven ancestral chromosomes went through the first WGD, followed by loss and fusion of one chromosome, thus generating 12 chromosomes. In the second WGD, 12 chromosomes went through duplication again, followed by a complex break and fusion, which leads to 27 modern chromosomes [42]. Conceivably, MYBs have also experienced an expansion during this process; we found that expansion of MYB TF genes mainly resulted from block duplication (R2R3-MYB subgroup) and tandem gene duplication (atypical-MYB) and might be associated with the adaptation of *P. patens* to terrestrial environments, as lineage-specific expansion of MYB TFs has been detected. In *P. patens*, ~8.6 (10/116)% MYBs (Figure 1) went through tandem gene duplication, which is similar to that observed in *A. thaliana* (~7%) [1]. In *A. thaliana*, the low tandem gene duplication has been suggested to be linked with highly conserved, housekeeping, or key regulatory gene families, whereas the medium- and high-tandem duplication classes have been suggested to be associated with pathogen defense or enzymatic functions [57]. Thus, low tandem gene duplication of MYB in *P. patens* probably reflects importance of MYBs as a key regulatory function. For instance, MpFGMYB in *M. polymorpha* and its orthologs in *A. thaliana* have been demonstrated to play a role in female gametophyte development [44,45]. Given the absence of sexual chromosome in *P. patens*, it would be of great interest in the future to explore whether orthologs (Pp3c6_23360 and Pp3c5_7650) of MpFGMYB play roles in archegonia development, because both Pp3c6_23360 and Pp3c5_7650 exhibit the highest expression in archegonia (Appendix A). 

*PpCCA1a* and *PpCCA1b* transcripts accumulate arrhythmically in continuous light, unlike their counterparts in angiosperms, which show sustained circadian rhythms with higher amplitudes [38]. The cause of this critical difference is unknown. Perhaps the different circadian behaviors of *P. patens* vs. angiosperms can (at least partly) be attributed to a dysfunction of the core clock components [38] and to the lack of some clock-related genes in *P. patens* [58]. It is also possible that tandem gene duplication contributed to the expansion and evolution of circadian clock components in plants by altering their expression patterns, domain organization, and subcellular localization, thus contributing to the flexibility of the circadian clock in bryophytes (Figure 1, Figure 2, Figure 3, Figure 4 and Figure 5). 

Although the roles of CCA1-like MYBs in regulating the circadian clock are well documented [34,35,36,38,39,59], their roles in stress responses are not. CCA1 regulates ROS (reactive oxygen species) homeostasis and oxidative stress responses by directly binding to the promoters of ROS genes and thereby coordinating time-dependent responses to oxidative stress [60]. Our results indicate that the expression of CCA1-like MYB genes can be induced by heat, suggesting that these genes are involved in the plant’s response to heat stress (Figure 5a,b). Moreover, some MYB genes, including CCA1-like MYBs, displayed tissue or developmental stage-specific expression patterns (for instance, archegonia-specific expression and expression during the chloronemata-to-caulonemata transition). The transition from chloronemata to caulonemata is regulated by environmental factors and the phytohormone auxin [61]. Interestingly, REVEILLE1, a MYB-like TF homologous to CCA1, regulates the expression of the auxin biosynthetic gene *YUCCA8*, thus functioning as an integrator linking the circadian clock to auxin pathways [5]. By contrast, R2R3-MYBs regulate auxin signaling by interacting with TFs, such as ARF (auxin response factors) [62] and TCP [63]. Therefore, it will be important to explore whether these stage-specific MYBs are responsible for the transition from chloronemata to caulonemata. 

The activity of transcription factors needs to be tightly controlled to regulate gene expression in response to intrinsic developmental cues or external environmental stimuli. This is achieved by regulation at multiple levels, including the transcriptional, posttranscriptional, translational, and posttranslational level [64,65]. Although the majority of TFs are nucleus localized, some TFs are not localized in the nucleus when they are initially synthesized. A well-characterized example is several members of the bZIP and NAC families. These TFs are kept as the dormant forms in the cytoplasm when they are synthesized or expressed as a membrane protein; upon stimulation, they are activated through proteolytic cleavage, thus releasing the active form that enters the nucleus and activates target genes [65]. In this study, we found that some MYBs (Pp3c24_8050, Pp3c4_2590, and Pp3c26_6730) are not localized in the nucleus. After close examination of these MYBTFs, we found that Pp3c24_8050 shares similarity with maMYB (At5g45420, 42% protein similarity), an endoplasmic reticulum membrane-bound TF (MTF) [64,66], which regulates root hair elongation [66]. Overall, our study lays the foundation for further exploration of the potential roles of MYBs in *P. patens*.

## 4. Materials and Methods

### 4.1. Plant Materials and Growth Conditions 

The Gransden wild-type strain of *P. patens* used for all experiments are provided by Prof. Mitsuyasu Hasebe. Protonemata and gametophores were cultured on solid BCDAT medium at 25 °C with a 16-h light/8-h dark cycle (60 to 80 μmol photons m^−2^ s^−1^), as previously described [67]. 

### 4.2. Heat Treatment, RNA Isolation, RNA-seq, and Data Analysis

Tissues from 6-day-old protonemata and 30-day-old gametophores were subjected to heat treatment as described previously [68] with minor modifications. Briefly, plates containing 6-day-old protonema and 30-day-old gametophore tissues were placed in an incubator at 40 °C for 18 h and transferred to normal growth conditions for 4 days for recovery. Following recovery, total RNA was isolated from the tissues using TRIzol reagent according to the manufacturer’s instructions and quantified with the NanoDrop 2000 Spectrophotometer (Thermo Fisher Scientific, Wilmington, USA). Library construction and Illumina sequencing were performed by the Beijing Genomics Institute (BGI) (Wuhan, China) in BGISEQ-500 platform as previously described [69]. Raw reads were cleaned by removing adapters and low-quality sequences with SOAPnuke (v1.5.2) software [70]. Clean reads were mapped to the *P. patens* reference genome v3.3 (http://www.phytozome.net/physcomitrella.php) using Bowtie2 (v2.2.5) [71]. The mapped reads were counted and the abundance was estimated using the RSEM method to obtain the FPKM and expected count [72]. The differential expression between two samples (control and heat stress) was identified based on the total mapping counts using the Bioconductor DEseq2 package [73]. The *p*-values were obtained from a differential gene transcription test. FDR (false discovery rate) was used to determine the threshold *p*-value. The threshold with log_2_fold-change (log_2_(treatment/ control)) cutoff of ≥2 and adjusted *p*-value of ≤0.05 were used to identify genes with significantly different levels of expression. The heat map was generated in R (http://www.r-project.org) using pheatmap package. 

### 4.3. Identification of MYB Genes in P. patens

The sequences of 125 R2R3-MYB proteins from *A. thaliana* [14] and the *P. patens* protein sequences were downloaded from the *Arabidopsis* Information Resource (TAIR10, https://www.arabidopsis.org/browse/genefamily/index.jsp) and Phytozome v12 (https://phytozome.jgi.doe.gov/pz/portal.html), respectively. The *P. patens* MYB proteins sequences were identified by all-to-all BLASTP using the 125 R2R3-MYB proteins from *Arabidopsis* as query sequences with an e-value cut-off of 1e–1. The resulting BLAST datasets of MYB sequences were designated as Dataset 1. Furthermore, *P. patens* MYBs were downloaded from PlantTFDB 4.0 (http://planttfdb.cbi.pku.edu.cn/index.php) and designated as Dataset 2. Datasets 1 and 2 were combined into a list of putative MYB proteins, which were further examined for consistency and for the presence of specific protein signatures using the InterProScan program (http://www.ebi.ac.uk/Tools/pfa/iprscan/) and InterPro database (http://www.ebi.ac.uk/interpro/). Ultimately, 116 *P. patens* MYB proteins were identified.

### 4.4. Multiple Sequence Alignment and Phylogenetic Analysis

The protein sequences of the 116 MYBs from *P. patens* were used to construct phylogenetic trees of the MYB superfamily. The sequences were aligned using MUSCLE3.831 [74] with default parameters, and poorly aligned positions were eliminated using the G-blocks server (http://molevol.cmima.csic.es/castresana/Gblocks_server.html). Using ProtTest version 3.2 [75], the LG+I+G model was found to be the best-fit model for amino acid evolution according to both the Akaike information criterion (AIC) and Bayesian information criterion (BIC). Phylogenetic analysis was performed using the maximum likelihood (ML) method with PHYML version 3.0, with optimization of the gamma shape parameter and 1000 bootstrap replicates. The phylogenetic trees were visualized using the iTOL web tool (http://itol.embl.de/index.shtml). The neighbor-joining (NJ) tree was constructed using MEGA5.2 [76]. The reliability of the internal branches was evaluated using 1000 bootstrap replicates.

### 4.5. Determining the Chromosomal Locations and Intron–Exon Structures of MYB Genes 

Information about the chromosome distribution (including the chromosome length and the starting positions of MYB genes) was retrieved from the Phytozome database and visualized using CIRCOS software [31]. The structures of MYB genes were visualized using the gene structure display server (GSDS) (http://gsds.cbi.pku.edu.cn/).

### 4.6. Conserved Motif Detection and Expression Pattern Analysis in P. patens

The conserved motifs in the MYB protein sequences were analyzed using the online tool Multiple Expectation maximization for Motif Elicitation (MEME, version 5.0.0, http://alternate.meme-suite.org/tools/meme) analysis tool with default parameters: Distribution of motifs, zero or one per sequence; maximum number of motifs to find, 20; minimum width of motif, 6; and maximum width of motif, 300. Only motifs with an e-value of 0.1 were retained for further analysis. Then, these conserved motifs were visualized through the weblogo server (http://weblogo.berkeley.edu/logo.cgi). Expression pattern data were retrieved from the BAR database (http://bar.utoronto.ca/efp_physcomitrella/cgi-bin/efpWeb.cgi).

### 4.7. Vector Construction and Subcellular Localization Analysis

Total RNA was isolated from two-week-old protonemata using an HiPure HP Plant RNA Mini Kit (R4165-02). Two micrograms of total RNA were used to synthesize first-strand cDNA for MYB gene cloning. To analyze the subcellular localization of the MYB proteins, full-length MYB CDS without stop codon from several representative MYB genes was cloned into the PM999 vector by one-step recombination cloning. Each CDS was fused in-frame to the N-terminus of the green fluorescent protein (GFP) coding sequence under the control of the CaMV 35S promoter. Primers used for PCR amplication of the full-length MYB CDS are listed in Appendix A. The resulting plasmids were transformed into protoplasts, which were incubated in darkness at 25 °C for 16 h and observed using a confocal microscope (Olympus FV1000, Tokyo, Japan). The subcellular localizations of the other MYB proteins were predicted using the Plant-PLoc server (http://www.csbio.sjtu.edu.cn/bioinf/plant-multi/).

## 5. Conclusions

In this study, a genome-wide identification and analysis of *MYB* genes encoded in the *Physcomitrella patens* genome were performed, which lead to 116 *MYB* genes being identified. These MYB proteins can be divided into four different types: The R2R3-MYB, R1R2R3-MYB, 4R-MYB subfamily, and the MYB-related subfamily. Interestingly, two types of N3R-MYB and NR-MYBs with a complicated gene structure appear to be novel MYB proteins found only in *P. patens.* In addition, we found that gene duplication contributes differentially to the expansion of a different MYB subgroup, of which block duplication is mainly responsible for the expansion of R2R3-MYB, whereas tandem gene duplication contributed greatly to the generation of the R1R2R3-, CCA1-like-, and N3R subgroup. Expression analysis of *P. patens MYB* genes in different developmental stages and stress conditions suggested that the MYB family has a wide expression profile, and plays important roles in stress responses. Our results provide a new insight into the potential role of MYB transcription factors in specific developmental transition and stress response in *P. patens*, thus laying the foundation for further functional investigation of *MYB* genes in early land plants.

## Figures and Tables

**Figure 1 ijms-21-00975-f001:**
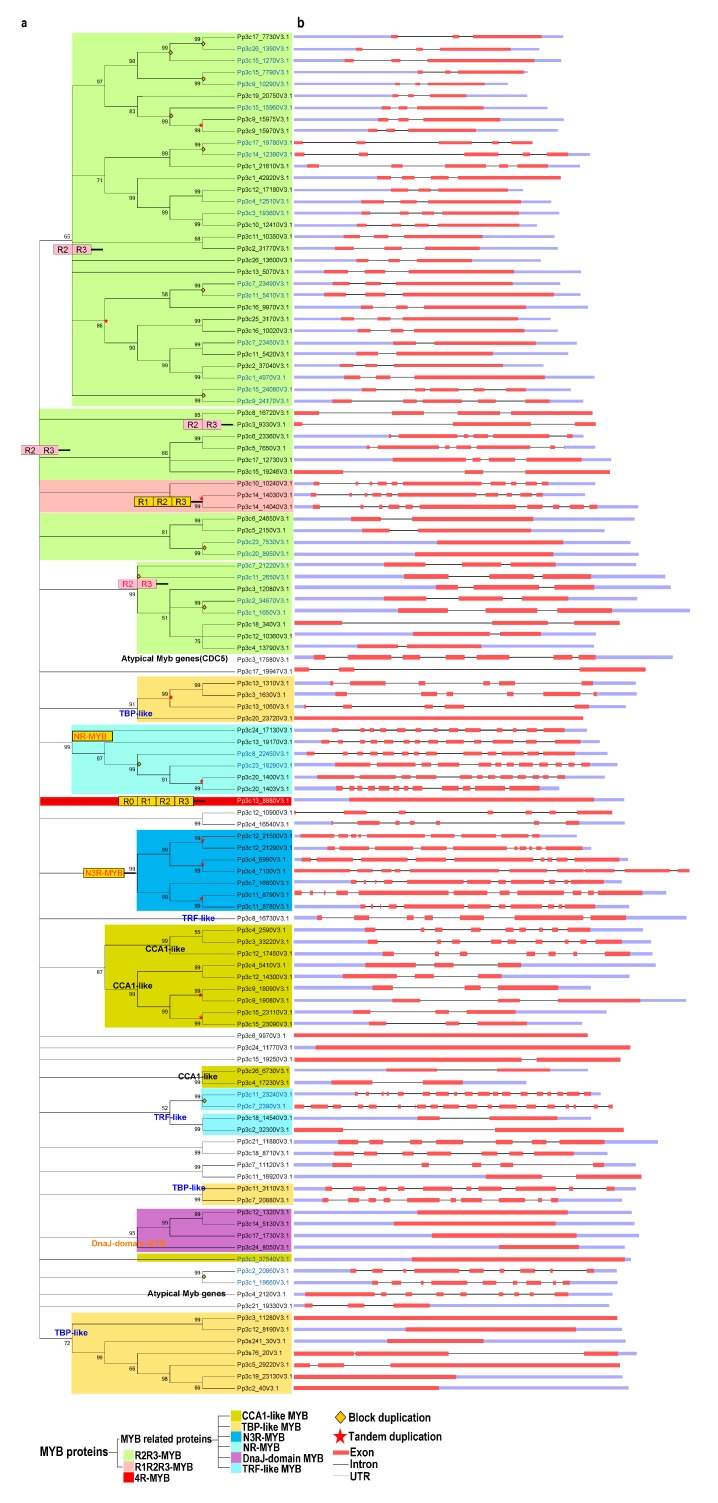
Phylogenetic and gene structure analyses of the MYB gene family in *Physcomitrella patens*. (**a**) The phylogenetic tree was constructed with MEGA 5.2 using the neighbor-joining (NJ) method with 1000 bootstrap replicates based on a multiple alignment of amino acid sequences of MYB genes from *P. patens*. The numbers above the branches represent the bootstrap support values (>50%) from 1000 replications. (**b**) A schematic diagram of the intron/exon structures of *MYB* genes from *P. patens*. Gene duplication events for gene family expansion are indicated by stars, of which stars colored in red and diamonds in orange represent tandem gene duplication events and block duplication events, respectively. Exons are indicated by a red rectangular box, introns are indicated by a blank line, and untranslated regions are indicated by a light blue rectangular box.

**Figure 2 ijms-21-00975-f002:**
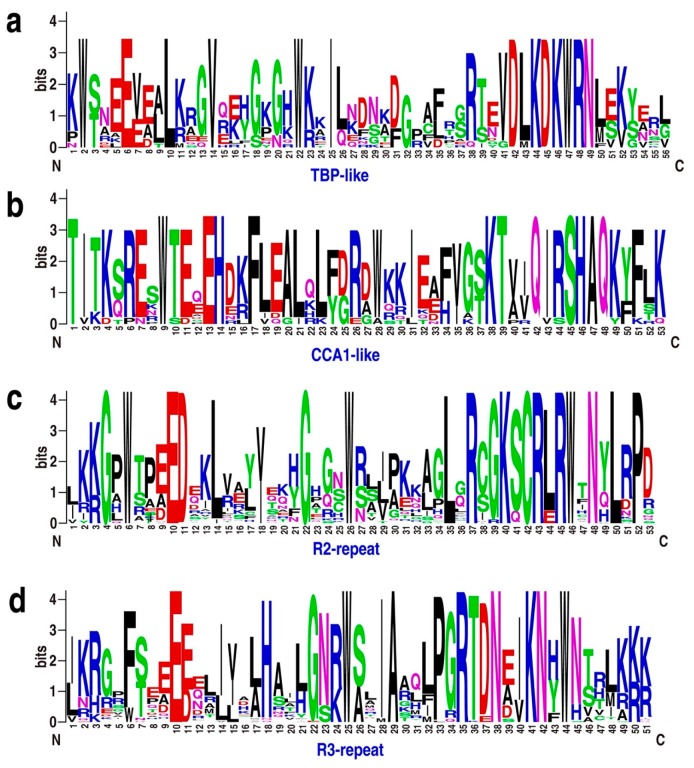
Sequence logos for the TBP-like (**a**), CCA1-like (**b**), R2 (**c**), and R3 (**d**) repeats found in *Physcomitrella patens* MYB proteins. Multiple alignment of 116 *P. patens* MYB domains was performed with MUSCLE software, and visualized by the web logo server. The bit score indicates the information content for each position in the sequence.

**Figure 3 ijms-21-00975-f003:**
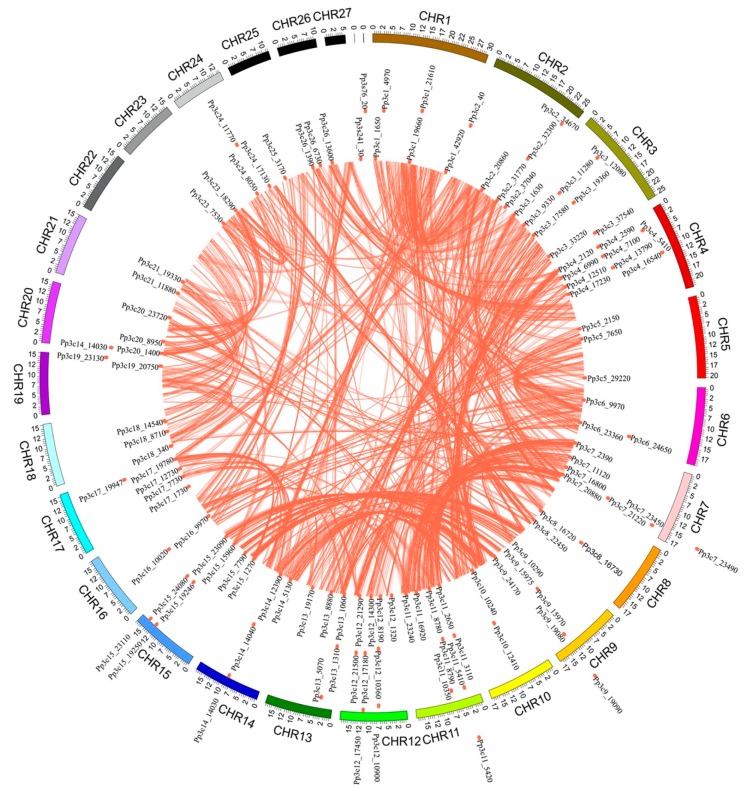
Chromosomal locations of MYB genes in the *Physcomitrella patens* genome. The *P. patens* chromosomes 1–27 are depicted in a circle with different colors. The terracotta bullets in front of the MYB genes indicate the position of MYB on the chromosome.

**Figure 4 ijms-21-00975-f004:**
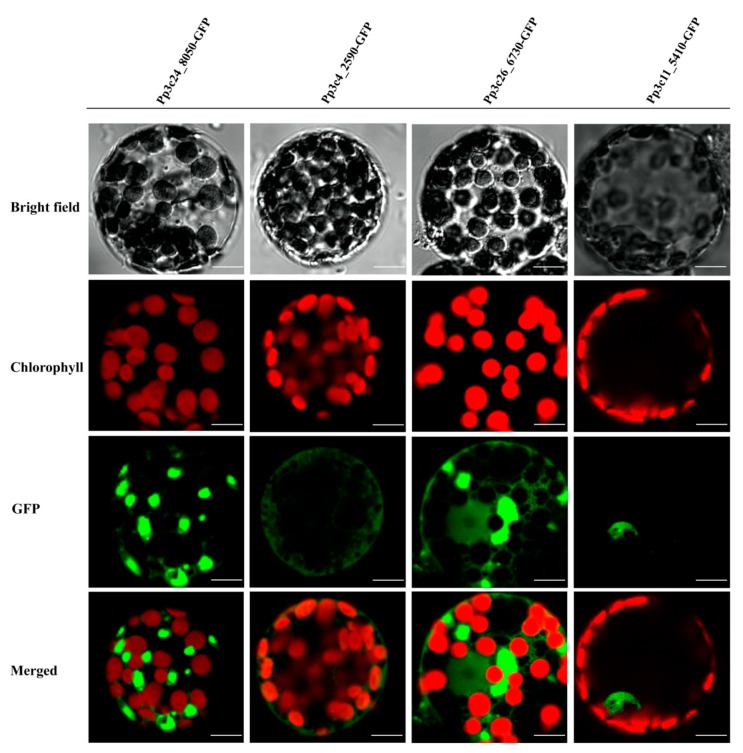
Subcellular localization of four different MYBs, Pp3c24_8050, Pp3c4_2590, Pp3c26_6730, and Pp3c11_5410, as indicated by green fluorescence. Subcellular localization of four different MYBs were analyzed, three of which (Pp3c24_8050, Pp3c4_2590, and Pp3c26_6730) are not localized to the nucleus, whereas 5410 (Pp3c11_5410) is localized in the nucleus as indicated by the green fluorescence. Scale bars, 10 μm.

**Figure 5 ijms-21-00975-f005:**
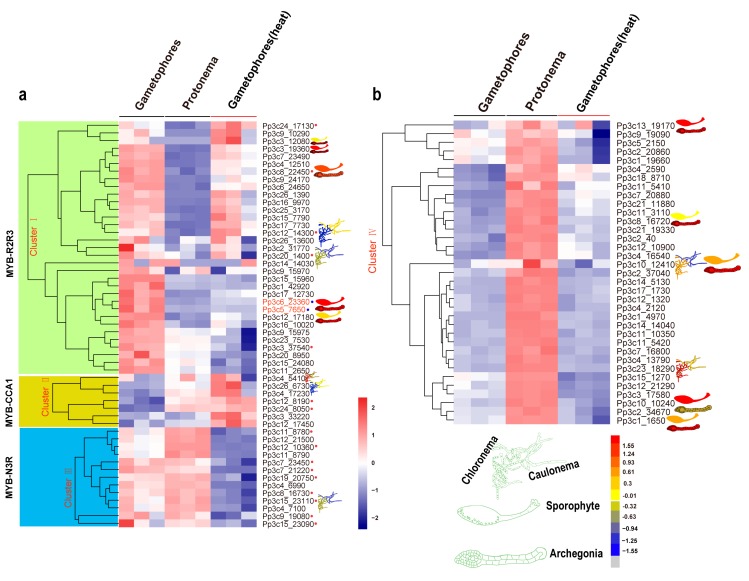
Expression profiles of *Physcomitrella patens MYB* genes at different developmental stages and in response to heat stress. Relative expression level of *MYB* genes in plant tissues and organs, in particular cell types, and different developmental stages were retrieved from the BAR database (http://bar.utoronto.ca). (**a**,**b**) The heat map shows the expression profiles of MYB genes in *P. patens* at the protonema and gametophore stages and at the gametophore stage after heat treatment. Each box in the heatmap represents a biological replicate. The color represents RPKM normalized log2-transformed value, blue means a downregulation gene, and brick means an upregulation gene. MYBs in the red circle belongs to different subgroups while the blue circle indicates orthologs of MpFGMYB in *P. patens*.

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
