# Peer review of "Genome-Wide Analysis of the MYB Transcription Factor Superfamily in Physcomitrella patens"

_ijms, 2020, doi:10.3390/ijms21030975_

Round 1

Reviewer 1 Report

The manuscript contains an analysis of the structure of an important family of transcription factors in an increasingly studied species. The results, although lacking significant biological information, can be useful for subsequent studies. My comments are essentially regarding discussion of other recent analyses of Physcomitrella genome. The paper of Lang et al, the paper of Perroud et al, both in Plant Journal, at least should be discussed.

Author Response

Reviewer1

The manuscript contains an analysis of the structure of an important family of transcription factors in an increasingly studied species. The results, although lacking significant biological information, can be useful for subsequent studies. My comments are essentially regarding discussion of other recent analyses of Physcomitrella genome. The paper of Lang et al, the paper of Perroud et al, both in Plant Journal, at least should be discussed.

Response: We have added corresponding references in introduction and discussion sections. Please see page 2, line72-75, page10, line252-256.

Reviewer 2 Report

The MS provides a comprehensive and systematic analysis of MYB-family proteins in P. patens and may be interesting for readers. I have the following comments and recommendations: 

1) Abstract: some less common abbreviations appear here without explanation - e.g., NR-MYB and N3R-MYB, CCA1-like, and their meaning is thus unclear.

2) Figure 1: The Figure legend includes designation of panels a and b, but these labels are not present in the Figure. Further, the mini-scheme below the phylogenetic tree is not legible. Besides the intron-exon structure (colours of introns and exons are not mentioned in the legend), It will be useful to depict also positions of myb-domains within a resulting protein molecule. 

3) Lanes 172-175 the authors state: These results suggest that R2R3-MYB genes are common in P. patens, as in other plants. Based on the classification used by Chen et al. (2000) [47] and our phylogenetic analysis, we further divided the 62 MYB-related proteins into several subgroups: CCA1-like, TBP-like, TRF-like, atypical MYB proteins, and other MYBs lacking of typical MYB characteristics.

First, the reference numbered as 47 does not correspond to Chen et al. (2000). Second, I believe that the classification of MYB genes/proteins should be explained directly in this MS (e.g., distinction between TBP-like and TRF-like is not clear from the current version). Does TBP-like subgroup correspond to the plant-specific TRB family of MYB-like proteins? Some recent article or review on telomeric proteins with myb-like domains should also be cited either in the Introduction or here.  

4) Figure 4 shows examples of various localisation patterns of four MYB proteins. However, it would be useful to show as examples (also) some representatives with a predicted nuclear function and also some with non-nuclear function (e.g., based on data on orthologues of A. thaliana counterparts), and discuss the corresponding localisation.

5) Paragraph 3.6: Please, compare the percentage of block and tandem duplications also between P. patens and respective subgroups in Arabidopsis thaliana or another land plant species. Compare the contribution of whole genome duplications between P. patens and A.thaliana.

6) Discussion: The first and the last paragraphs of the Discussion should be merged (without redundancies) and used as the first chapter of the Discussion. 

Author Response

Reviewer2

1) Abstract: some less common abbreviations appear here without explanation - e.g., NR-MYB and N3R-MYB, CCA1-like, and their meaning is thus unclear.

Response: We have added the explanation for these abbreviations, Please see page1, line18.

2) Figure 1: The Figure legend includes designation of panels a and b, but these labels are not present in the Figure. Further, the mini-scheme below the phylogenetic tree is not legible. Besides the intron-exon structure (colours of introns and exons are not mentioned in the legend), It will be useful to depict also positions of myb-domains within a resulting protein molecule. 

Response: We have revised figure1 as reviewer suggested. Please see page5, line117-118.

3) Lanes 172-175 the authors state: These results suggest that R2R3-MYB genes are common in P. patens, as in other plants. Based on the classification used by Chen et al. (2000) [47] and our phylogenetic analysis, we further divided the 62 MYB-related proteins into several subgroups: CCA1-like, TBP-like, TRF-like, atypical MYB proteins, and other MYBs lacking of typical MYB characteristics. First, the reference numbered as 47 does not correspond to Chen et al. (2000). Second, I believe that the classification of MYB genes/proteins should be explained directly in this MS (e.g., distinction between TBP-like and TRF-like is not clear from the current version). Does TBP-like subgroup correspond to the plant-specific TRB family of MYB-like proteins? Some recent article or review on telomeric proteins with myb-like domains should also be cited either in the Introduction or here.  

Response: We have changed classification used by Chen et al. (2000) [47] into classification used by Chen et al. (2006) [47]. Please see page5, line123.

4) Figure 4shows examples of various localization patterns of four MYB proteins. However, it would be useful to show as examples (also) some representatives with a predicted nuclear function and also some with non-nuclear function (e.g., based on data on orthologues of A. thalianacounterparts), and discuss the corresponding localization.

Response: We have added corresponding contents in discussion. Please see page 11, line 291-301.

5)Paragraph 3.6: Please, compare the percentage of block and tandem duplications also between P. patensand respective subgroups in Arabidopsis thaliana or another land plant species. Compare the contribution of whole genome duplications between P. patens and A.thaliana.

Response: We have added corresponding contents as reviewers suggested. Please see page10, line245-251.

6) Discussion:The first and the last paragraphs of the Discussion should be merged (without redundancies) and used as the first chapter of the Discussion. 

Response: We have merged the first and the last paragraphs of the Discussion as reviewers suggested. Please see page10, line245-251.

Reviewer 3 Report

In this article, Pu and colleagues report a comprehensive analysis of MYB transcription factors in Physcomitrella patens. Combining a blast-search approach and data from PlantTFDB, they identified a total of 116 MYB proteins within the genome assembly, from which they determined their chromosomal location and performed a phylogenetic analysis. Then, they investigated the structure of these MYB factors and their cellular localisation using both protoplast transformation and bioinformatics predictions. They also studied their gene expression using RNA-seq analyses, including heat stress conditions in their experiments.

Despite the study was well designed and the article is well written (I have particularly appreciated the introduction), I have shortcomings that the authors should consider to improve their manuscript.

Introduction

L57: the authors should group references 29 to 31.

L61: the sentence should not initiate a new paragraph.

L77: the authors should develop “little is known about these gene family”. For example, how many MYB factors were previously identified in P. patens in studies from Du et al, 2015?

Materials and Methods

L88, the authors explain they subjected tissues from protonema and gametophores to heat treatment but provided only results for gametophores (Figure 3).

Overall, information concerning bioinformatics analyses are too poor. For example, L94 there is no information how are obtained clean reads. Please cite tools used to filter low-quality sequences, adaptors, etc and provide parameters used to define a sequence as being of good quality. In general, I advise the authors to provide a reference (or a link to a website if the tool is not published) and the version number for each software they used.

L96 I don’t know whether replicates are used in the RNA-seq analysis.

L103 I don’t understand why the authors used BlastP while they compare protein sequences to a nucleotide database (the P. patens genome). Did they mean tblastn or did they compare MYB factor sequences to translated CDS obtained from the genome?

Results

L149-150: same as just above with BlastP, it is not clear for me.

L221-L222: the authors should compare the location of MYB genes within the genome to a random distribution to state that MYB genes are dispersed non-uniformly.

Figure3: It is not indicated what mean orange lines connecting chromosomes in the Circos Figure. Also, the IDs of some MYB overlap with chromosomes. The figure could be improved.

Figure 5: why the figure is separated in a and b?

In the heat map, I don’t understand what mean the 3 boxes for Gametophores, Protonema and Gametophores(heat). Different developmental stages? Replicates?

L275: “were induced”: The authors should provide p-values to support their statements. Ditto L277 “were less strongly induced”.

Author Response

Reviewer3

Introduction, L57: the authors should group references 29 to 31.

Response: We have grouped references into 29-31. Please see page2, line60.

L61: the sentence should not initiate a new paragraph.

Response: We have combined this section. Please see page2, line 60-68.

L77: the authors should develop “little is known about these gene family”. For example, how many MYB factors were previously identified in P. patens in studies from Du et al, 2015?

Response: We have added the contents as reviewer suggested. Please see page2, line 88-92.

Materials and Methods, L88, the authors explain they subjected tissues from protonema and gametophores to heat treatment but provided only results for gametophores (Figure 3). Overall, information concerning bioinformatics analyses are too poor. For example, L94 there is no information how are obtained clean reads. Please cite tools used to filter low-quality sequences, adaptors, etc and provide parameters used to define a sequence as being of good quality. In general, I advise the authors to provide a reference (or a link to a website if the tool is not published) and the version number for each software they used.

Response: We have added details of analysis of RNA-Seq, Please see page11, line 300-310.

L96 I don’t know whether replicates are used in the RNA-seq analysis.

Response: We have three biological replicates each sample.

L103 I don’t understand why the authors used BlastP while they compare protein sequences to a nucleotide database (theP. patensgenome). Did they mean tblastn or did they compare MYB factor sequences to translated CDS obtained from the genome?

Response: We have rectified our description, Please see page11, line 317.

Results, L149-150: same as just above with BlastP, it is not clear for me.

Response: We have rectified our description, Please see page11, line 317.

L221-L222: the authors should compare the location of MYB genes within the genome to a random distribution to state that MYB genes are dispersed non-uniformly.

Response: We have rectified our manuscript, Please see page6-7, line 166-169.

Figure3: It is not indicated what mean orange lines connecting chromosomes in the Circos Figure. Also, the IDs of some MYB overlap with chromosomes. The figure could be improved.

Response: Linked lines in Circos figure indicate syntenic regions of all P. patens genes. We have revised the figure, please see page7, line169-171.

Figure 5: why the figure is separated in a and b?

Response: Because figure is too long, it will become clear when separated into two parts.

In the heat map, I don’t understand what mean the 3 boxes for Gametophores, Protonema and Gametophores(heat). Different developmental stages? Replicates?

Response: The 3 box represents different replicates, we have added a detailed description in figure 5. Please see page 9, line 199-205.

L275: “were induced”: The authors should providep-valuesto support their statements. Ditto L277 “were less strongly induced”.

Response:The heatmap was generated based on the expression fold changes between treatments and control conditions; p-values were calculated for all comparisons of transcript changes under different conditions (See method and materials). Therefore, the statements in these sentences are statistically-based. In addition, these statements are generally interpreting the results of heatmap, not overemphasizing the statistical significance.We think that the statements are unambiguous and logically accurate.

Round 2

Reviewer 2 Report

Although the authors improved the MS in the revised version, it would still benefit from improvement of the results presentation.

Specifically, this concerns figures:

Figure 1: the explanatory tree in the bottom is still not legible

Figure 3 legend does not mention the meaning of terracota bullets in front of most MYB genes. 

Figure 5: pictograms in the bottom are hardly legible, signs (probably) "Caulonema" and "Chloronema" are not legible in normal size at all. These pictograms are not suitable at all for the use besides expression data - they are too complicated to be clearly distinguishable in such size.

Author Response

Reviewer2

Although the authors improved the MS in the revised version, it would still benefit from improvement of the results presentation.

Specifically, this concerns figures:

1.Figure 1: the explanatory tree in the bottom is still not legible

Response:We have enlarged the figure. We also provided the high-resolution files of figure1 in PDF and ai format. Please see figure folder.

2.Figure 3 legend does not mention the meaning of terracota bullets in front of most MYB genes. 

Response: We have added description in the figure legend. The terracota bullets in front of MYB genes indicates the position of MYB on chromosome. Please see page7, line188-189.

3.Figure 5: pictograms in the bottom are hardly legible, signs (probably) "Caulonema" and "Chloronema" are not legible in normal size at all. These pictograms are not suitable at all for the use besides expression data - they are too complicated to be clearly distinguishable in such size.

Response: We have enlarged the figure. We also provide the high-resolution files of figure5 in PDF and ai format. Please see figure folder.

Reviewer 3 Report

All the points I raised in my previous review have been considered by the authors, but for some of these shortcomings the authors' response remains partial.

Line 318, the authors do specify they used SOAPnuke to clean the Illumina reads but they omit to precise the version. Moreover they should give the reference instead of the web link (Chen Y, et al. SOAPnuke: a MapReduce acceleration-supported software for integrated quality control and preprocessing of high-throughput sequencing data. Gigascience. 2018 Jan 1;7(1):1-6).

L316 illumina should be written with a capital letter.

L330, I am not satisfied with the answer given concerning the blast analysis:

1) It seems to me that “The P. patens proteomic MYB proteins sequences” should be written “The P. patens MYB protein sequences”.

2) I understand that the query is the set of 125 MYB proteins from A. thaliana but the authors should specify the protein database (subject) they used (translated CDS obtained from genome annotations?). The authors indicate “The sequences of 125 R2R3-MYB proteins from A. thaliana and the P. patens genome sequence were downloaded…”, which suggests they performed a tblasn and not a blastp search.

Figure 3: the authors answered my question concerning orange lines in the Circos plot that they indicate syntenic regions of all P. patens genes. However, I wonder if it is relevant to give the syntheny information in the figure, since it is not commented in the manuscript. If it seems relevant to the authors, they should then complete the figure caption.

Author Response

Reviewer3

All the points I raised in my previous review have been considered by the authors, but for some of these shortcomings the authors' response remains partial.

Line 318, the authors do specify they used SOAPnuke to clean the Illumina reads but they omit to precise the version. Moreover they should give the reference instead of the web link (Chen Y, et al. SOAPnuke: a MapReduce acceleration-supported software for integrated quality control and preprocessing of high-throughput sequencing data. Gigascience. 2018 Jan 1;7(1):1-6).

Response: We have added the reference in manuscript as the reviewer suggested.

L316 illumina should be written with a capital letter.

Response: We have revised our manuscript as the reviewer suggested. Please see page12, line343-346.

L330, I am not satisfied with the answer given concerning the blast analysis:

1) It seems to me that “The P. patens proteomic MYB proteins sequences” should be written “The P. patens MYB protein sequences”.

Response: We have deleted the word “proteomic” as the reviewer suggested. Please see page12, line360.

2) I understand that the query is the set of 125 MYB proteins from A. thaliana but the authors should specify the protein database (subject) they used (translated CDS obtained from genome annotations?). The authors indicate “The sequences of 125 R2R3-MYB proteins from A. thalianaand the P. patens genome sequence were downloaded…”, which suggests they performed a tblasn and not a blastp search.

Response: We have revised our manuscript as the reviewer suggested. Specifically, 125 R2R3-MYB MYB proteins from A. thaliana were downloaded from this website (https://www.arabidopsis.org/browse/genefamily/MYB.jsp). Protein sequence from P. patens was downloaded from Phytozome v12 (https://phytozome.jgi.doe.gov/pz/portal.html) database, which requires registration before download. Therefore, we do perfomed BLASTP search. Please see page12, line356-362.

Figure 3: the authors answered my question concerning orange lines in the Circos plot that they indicate syntenic regions of all P. patens genes. However, I wonder if it is relevant to give the syntheny information in the figure, since it is not commented in the manuscript. If it seems relevant to the authors, they should then complete the figure caption.

Response: We have revised our manuscript as the reviewer suggested. We do perform syntenic regions of all P. patens genes and mark the position of MYB on chromosome. Please see page7, line188-189.